# Understanding Distribution Alignment Through Category Separability In An Infant-Inspired Domain Adaptation Task

## Abstract

We introduce a novel distribution shift, called the VI-Shift, that mimics the trade-off between object instances and viewpoints in the visual experience of infants. Motivated by findings in infant learning literature, we study this problem through the lens of domain adaptation, but without ImageNet pretraining. We show that the performances of two classic domain adaptation methods, Joint Adaptation Network (JAN) and Domain Adversarial Neural Networks (DANN), deteriorate without ImageNet pretraining. We hypothesize that the separability of source and target category clusters in the feature space plays a crucial role in the effectiveness of JAN. So, we propose 3 metrics to measure category separability and demonstrate that target separability in the pretrained network is strongly correlated with downstream JAN and DANN accuracy. Further, we propose two novel loss functions that increase target separability during pretraining by aligning the distribution of within-domain pairwise distances between the source and target distributions. Our experiments show that the application of these loss functions modestly improves downstream accuracy on unseen images from the target dataset.

## 1 Introduction

Imagine an infant engaged in play with their mother. As the infant uses their budding grasping abilities to interact with their toy cars, their mother repeats the word 'car' to them. Over time, the infant learns to associate the word 'car' with the physical toy cars before them (Pereira et al., 2014).

Infants' visual experiences are characterized by extended bouts of experience with a small number of familiar objects (e.g., toy ducks at home), with a large number of rarer exposures to less familiar objects (e.g., real ducks at the park). This pattern of exposure to instances of a particular category yields a long-tailed distribution, where some instances (e.g. their toys/household objects) are seen very frequently, while most instances (e.g. objects they see outdoors) are seen more rarely: **(1)** The *head* of the distribution is rich in the distribution of viewpoints, i.e. *viewpoint-dominated* (**VD**), while **(2)** the *tail* of the distribution is rich in the number of different category instances, i.e. *instance-dominated* (**ID**).

Infants learn about categories by linking heard words to the objects they see. However, in natural interactions with parents, the share of object words tends to be quite low (Stärk et al., 2022). Additionally, visual scenes for infants are often cluttered (Clerkin et al., 2017); there is no clear object that a heard word refers to. Learning with such ambiguity and noisy signals can be difficult. In contrast, joint play experiences with parents offers more straightforward opportunities for category learning. When playing with an object, an infant's visual scene is dominated by the object they are holding (Smith et al., 2011). Armed with clear visual targets, infants use heard nouns more efficiently to learn object names (Suanda et al., 2019; Pereira et al., 2014). Further, these sessions often generate more verbal inputs from parents (Tamis-LeMonda et al., 2017), making these experiences potentially better learning experiences for infants.

This work is motivated by the contrasts between the **VD** experience during object play and **ID** experience otherwise; both in distributions of visual experience and in the presence of clear learning signals in the form of heard words. Specifically, we ask the following question: *To what extent is it possible to successfully classify unlabeled ID images by learning directly from **labeled VD** and **unlabeled ID** images?*

This question is similar to the domain adaptation (DA) framework (Ben-David et al., 2010) in ML; we use the VD dataset as the source and the ID dataset as the target. Existing DA methods (Long et al., 2015; French et al., 2017) rely on ImageNet (Deng et al., 2009) pretraining; they leverage high-quality features that comes with supervised training on massive, labeled datasets. This is not developmentally plausible, as infants do not learn from massive, labeled datasets. Instead, we investigate learning features directly from VD and ID datasets. We argue that: in learning directly from VD and ID distributions, learners can leverage *cross-distribution* learning signals that enforce consistency between the two distributions. As existing methods rely on ImageNet pretraining, there is a dearth of models that use *cross-distribution* signals to learn features directly from the task data.

Our contributions are:

- We introduce a novel distribution shift, called the VI-Shift, and investigate learning under this distribution shift using the domain adaptation framework. Consistent with our developmental motivation, we evaluate two classic yet effective DA methods, Domain Adversarial Neural Network (DANN) (Ganin et al., 2016) and Joint Adaptation Network (JAN) (Long et al., 2017) on the VI-Shift without ImageNet pretraining and show that this causes degradation in performance.
- We investigate how separability of category clusters in the pretrained network affects downstream DA evaluation. To this end, we propose three metrics to measure the separability of category clusters. Using these metrics, we show that DA accuracies using both DANN and JAN on the target dataset are strongly correlated with the separability of target clusters in the pretrained network.
- We propose two Maximum Mean Discrepancy (Gretton et al., 2012) based loss functions for improving the separability of target categories during pretraining. These losses align the distributions of pairwise image distances across the two datasets. We apply these losses in conjunction with contrastive learning signals. Our results show that the application of these losses leads to both improved performance and increased category separability in the feature space.

## 2    OUR APPROACH

### 2.1    VI-SHIFT

Visual experience during object play is *viewpoint-dominated* with a relatively small number of objects. Developmental psychologists have observed that such experiences drive visual learning in infants (Yurovsky et al., 2013; Clerkin et al., 2017). Further, parents often provide object labels (Suanda et al., 2019), which can provide straightforward opportunities for category learning. In contrast, most other object experiences are *instance-dominated*, where available labels can be sparse and noisy (Clerkin et al., 2017; Stärk et al., 2022). We call this the **VI-Shift**. We instantiate VI-Shift using the Toybox (Wang et al., 2018) dataset and by curating a category-matched ID dataset from ImageNet (Deng et al., 2009) and MS-COCO (Lin et al., 2014).

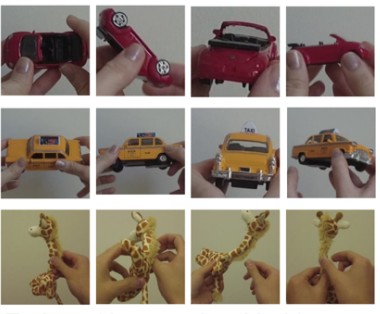
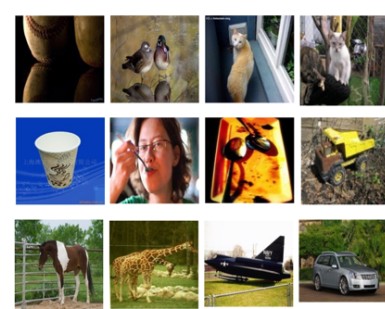

**Toybox:** 12 categories, 30 objects per category, many viewpoints per object

**IN-12:** 12 categories, many objects per category, 1 viewpoint per object

Figure 1: The Toybox → IN-12 distribution shift problem. The distribution shift mimics the distribution shift encountered in an infant's visual experience.

**Toybox dataset**    The Toybox dataset contains short egocentric videos of objects being manipulated in different ways. The dataset contains 360 objects from 12 categories; the categories in the dataset can be grouped into 3 super-categories: vehicles (airplanes, cars, helicopters, trucks), animals (cat, duck, giraffe, horse) and household objects (balls, cups, mugs, spoons). We use the Toybox dataset

in our experiments for the following reasons: (1) Toybox categories correspond to early learned nouns among children in the US (Fenson et al., 2007) and increases the developmental relevance of the considered categories. (2) The dataset contains videos depicting a wide variety of controlled, such as rotation, and random object manipulations (*hodgepodge*) leading to a large variety of viewpoints for each object.

**IN-12 dataset** To create a category matched dataset for the Toybox dataset, we curated the IN-12 dataset from the ImageNet (Deng et al., 2009) and MS-COCO (Lin et al., 2014) datasets. First, we manually extracted all ImageNet classes corresponding to the 12 Toybox categories. From among these candidate classes, we select a few synsets which describe the category at a general level (e.g. car vs police car). From these chosen synsets, we randomly select 1600 images per class while ensuring that each candidate synset contributed the same number of images. The entire list of the synsets are presented in Fig 7 in the Appendix. For the giraffe and helicopter categories, we extracted additional images from the MS-COCO dataset because the ImageNet synsets did not contain sufficient number of images. Fig 1 shows example images depicting this task.

## 2.2 COMPARISON WITH OTHER DISTRIBUTION-SHIFT DATASETS

Handling distribution shifts is an important problem; otherwise, models fail to generalize when the test distribution differs from the training distribution (Torralba & Efros, 2011). Several tasks have been proposed considering different kinds of distribution shift: for image classification, tasks like domain adaptation (Ben-David et al., 2010) and domain generalization (Blanchard et al., 2011) provide different frameworks for handling the distribution shift problem. Different datasets have been proposed that to handle distribution shift: Office-31 (Saenko et al., 2010), Office-Caltech (Gong et al., 2012), PACS (Li et al., 2017), Office-Home (Venkateswara et al., 2017), DomainNet (Peng et al., 2019) and ImageNet-R (Hendrycks et al., 2021) are some popular datasets used for distribution shift problems. However, the VI-Shift is different from these existing distribution shifts. While these datasets handle a variety of distribution shifts such as changes in camera source (Office-31, Office-CalTech) and image rendition styles (PACS, DomainNet, ImageNet-R), the different domains within these datasets are still Instance-Dominated. These datasets do not have a data distribution that would be called Viewpoint-Dominated.

The VisDA-2017c (Peng et al., 2017) shift is similar to the VI-Shift problem; however, the Viewpoint-Dominated dataset in the VisDA-2017c dataset consists of 2d renderings of simple 3d models. These images do not capture texture and color of real-world objects.

## 2.3 EXPERIMENTAL SETTING: DOMAIN ADAPTATION

Domain Adaptation is a popular framework for addressing distribution shifts between training and test data. The theory of domain adaptation suggests that good performance on the test domain can be achieved by jointly optimizing a source domain error and the divergence between the two domains (Ben-David et al., 2010). In this work, we use two classic DA methods, Joint Adaptation Network (JAN) (Long et al., 2017) and Domain Adversarial Neural Network (DANN) (Ganin et al., 2016). These methods take a complementary approach towards reducing the divergence loss: while JAN directly minimizes an explicit alignment loss, DANN takes an adversarial approach that makes determining the domain label for datapoints difficult.

**Joint Adaptation Network (JAN)** JAN jointly optimizes a classification loss on the source dataset and alignment loss on the distribution of target features with the source features. The JAN alignment loss, called the JMMD loss is based on the Maximum Mean Discrepancy (MMD) loss (Gretton et al., 2012) which measures the distance between two distributions as the distance between their mean embeddings in a RKHS. Given two datasets $X_s$ and $X_t$ with $n_s = |X_s|$ and $n_t = |X_t|$, the MMD loss is defined as:

$$l_{mmd} = \frac{1}{n_s^2} \sum_{x \in X_s} \sum_{y \in X_s} k(x, y) + \frac{1}{n_t^2} \sum_{x \in X_t} \sum_{y \in X_t} k(x, y) - \frac{2}{n_s n_t} \sum_{x \in X_s} \sum_{x \in X_t} k(x, y)$$

**Domain Adversarial Neural Network (DANN)** DANN adopts an adversarial signal to minimize the domain divergence. Specifically, they augment the source classification loss with an additional loss for predicting the domain label for datapoints. Ganin et al. (2016) use a Gradient Reversal Layer

to encourage gradient updates that make determining the domain label difficult, thereby reducing the divergence.

## 2.4 JAN PERFORMANCE DETERIORATES WITHOUT ILSVRC PRETRAINING

### 2.4.1 PRETRAINING SCHEMES

We use the following pretraining schemes and evaluate how JAN performs under these varying settings:

1. Random Initialization: We initialize the network with random weights and apply JAN directly. This is expected to be the most difficult condition.
2. Toybox Supervised: We pretrain the network using supervised learning on the Toybox dataset. This condition is also likely to be difficult for JAN because the pretrained network has not been exposed to any *instance-dominated* dataset.
3. IN-12 SSL: We use self-supervised learning (SSL) to pretrain the network using IN-12 images. Specifically, we use the Decoupled Contrastive Loss (DCL) (Yeh et al., 2022) for this purpose.
4. IN-12 Supervised: In this condition, we pretrain the network directly on the target IN-12 dataset using supervised learning. We expect JAN to perform well with this initialization.
5. Joint Supervised: Here, we pretrain the network by training it jointly on both Toybox and IN-12 dataset using supervised learning. In this setting too, we expect JAN to do well.
6. Joint Supervised-24: This is a variant of the previous setting; we distinguish between the two datasets so the network is presented with Toybox cars and IN-12 cars as two separate categories.
7. ILSVRC pretraining: This is the default experimental setting for domain adaptation experiments and we expect JAN to perform well under this condition.

### 2.4.2 EXPERIMENT DETAILS

We use a ResNet-18 He et al. (2016) backbone in our experiments. For the pretraining methods that require training, we initialize the network with the Xavier initialization (Glorot & Bengio, 2010) and train the networks from scratch on each of the different experimental settings. We use the Adam optimizer (Kingma & Ba, 2014) for training the network. During training, we linearly increase the learning rate for the first 2 epochs of training and then decay the learning rate using a cosine decay schedule (Loshchilov & Hutter, 2016) without any restarts. [1] Further experimental details for JAN and DANN are provided in the appendix.

### 2.4.3 RESULTS

Fig 2 shows the accuracies obtained by JAN and DANN evaluation under different pretraining conditions. We see that performance of both JAN and DANN degrades without ImageNet pretraining. This drop is more significant for the developmentally plausible pretraining (green shaded background). These results suggest that DA methods like JAN and DANN rely significantly on the quality of pretrained features. It is to be noted that DA performance without ImageNet pretraining lags behind the performance using linear evaluation on these same models; JAN and DANN are unable to learn good classifiers even when they exist. The full results for linear and DA evaluation are available in Tables 3, and 4 in the Appendix.

## 3 CATEGORY SEPARABILITY IN THE FEATURE SPACE AND ITS RELATION TO DOWNSTREAM JAN PERFORMANCE

### 3.1 DOMAIN ALIGNMENT AND SEPARABILITY

The previous results show that JAN and DANN require high-quality features from pretraining for good performance. *What factors of the pretrained feature space are beneficial for DA performance?* The answer to this question would help us design a pretraining method using only Toybox and IN-12 data. We hypothesize that the separability of source and target categories in the feature space plays an important role for these methods. DANN and JAN learn by jointly optimizing a source loss and a divergence loss (Ben-David et al., 2010); the divergence loss reduces the *global* shift between the two distributions. The effectiveness of this divergence loss depends on how able it is at aligning *category-consistent* regions (Toybox cars with IN-12 cars) of the feature space for the two

---

[1]Our code can be accessed at anonymized gdrive link.

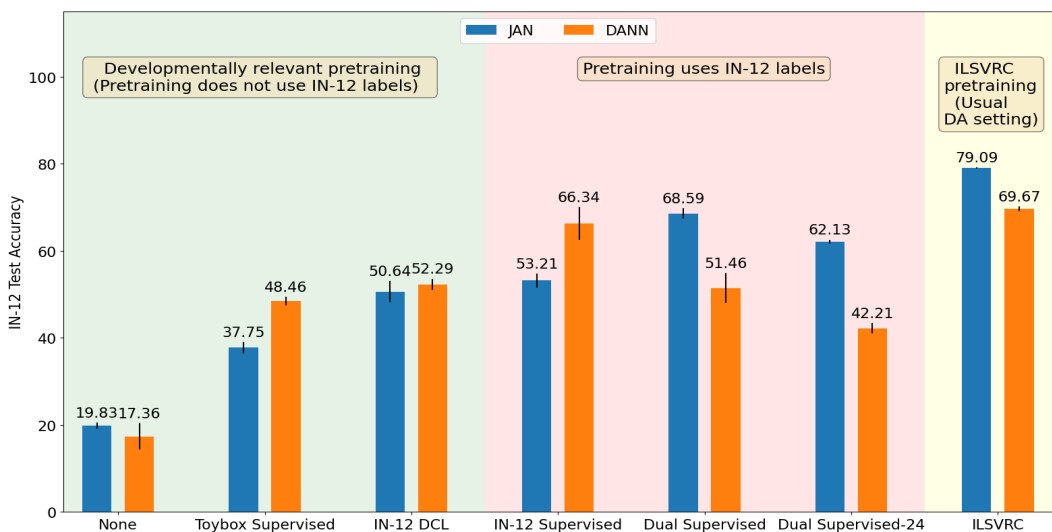

Figure 2: Accuracy on IN-12 test images using different pretraining schemes with JAN and DANN

distributions. We argue that high separability of category clusters in the pretrained networks aids in aligning such *category-consistent regions* using their respective divergence losses. To test this hypothesis, we propose 3 metrics for measuring separability between category clusters in the source and target datasets. Using these metrics, we show that *downstream test accuracies with DANN and JAN are highly correlated with the separability of target clusters and to a limited extent, with the separability of source clusters*.

## 3.2 CATEGORY DISTRIBUTION MODELING

We model the group of images in category as a probabilistic distribution. To do this, we utilize 2d embeddings obtained from running UMAP on the dataset.

### 3.2.1 UMAP EMBEDDINGS

We use UMAP (McInnes et al., 2018) to obtain 2D features for each datapoint. UMAP is a popular dimensionality reduction tool; it uses a manifold learning approach to find low-dimensional embeddings that are faithful to local structure in high dimensions. Two details are important here: (i) We run UMAP jointly using both datasets; this enables UMAP to learn a reducer that takes into account the joint structure of the two datasets. (ii) We do not use image labels during UMAP. This ensures that the low-dimensional embeddings are built only from the geometric structure of the datapoints.

### 3.2.2 OUTLIER REMOVAL

To reduce the impact of outliers during probabilistic modeling, we remove them from the 2D datapoints for each category by adopting a non-parametric approach (Wilkinson et al., 2005) based on minimum spanning trees (MSTs). Specifically, we construct an MST for every category and remove some of the longer edges leading to disconnected components. Then, we discard the components which have size less than 5. To remove edges, we apply an adaptive threshold, $\tau = q_{97.5} + 1.5 * (q_{97.5} - q_{2.5})$, where $q_{97.5}$ and $q_{2.5}$ are the 97.50-th and 2.50-th quantile edge lengths. Any edge longer than $\tau$ is dropped from the MST. This ensures that less than 5% of the edges are dropped; in practice, we find that very few are dropped.

### 3.2.3 DISTRIBUTION ESTIMATION

We adopt two approaches for modeling the probability distribution underlying each category, one non-parametric and one parametric. As the non-parametric method, we use Kernel Density Estimation and as the parametric method, we use Gaussian Mixture models.

**Kernel Density Estimation (KDE)** Kernel density estimation (Davis et al., 2011) is a technique for non-parametric probability distribution estimation; it does this by placing a smooth kernel at the

Figure 3: Distribution modeling steps for 3 pretraining conditions for the IN-12 data points. Starting from UMAP embeddings for a category, firstly, we remove outliers; few points are lost in this step. Secondly, we use KDE or GMM modeling to fit a probability distribution to category samples. In the Random Initialization case, the category samples are highly overlapping with each other, which is seen in the similar KDE and GMM heatmaps. The two categories are quite distinct in the IN-12 Supervised model, while the category separation is intermediate in the IN-12 DCL network.

location of each datapoint. Given samples $X^c = \{x_1^c, x_2^c, \dots x_n^c\}$ from category $c$, we estimate the probability distribution $p_{X^c}$ underlying $X^c$ as:

$$p_{X^c}(x; h) = \frac{1}{n} \sum_{i \in [n]} K\left(\frac{x - x_i^c}{h}\right)$$

where $K$ is the Gaussian kernel and $h$ is the kernel bandwidth parameter. We select the bandwidth value by 5-fold cross-validation on $X^c$.

**Gaussian Mixture Models (GMM)** Gaussian Mixture models, on the other hand, learns a probabilistic model composed of one or more parameterized Gaussian distributions. It is helpful when sub-populations of the data belong to different distributions. Given samples $X^c$ from category $c$, we model the distribution $p_{X^c}$ as:

$$p_{X^c}(x; \phi, \mu, \Sigma) = \sum_{i \in [K]} \phi_i \mathcal{N}(x; \mu_i, \Sigma_i)$$

We use the number of components, their means, variances and relative size as initial estimates for estimating the GMM parameters $\phi, \mu$ and $\Sigma$.

Fig 3 shows the distribution modeling steps for 3 example pretraining schemes for the IN-12 data points. In the Random Initialization case, the category samples are highly overlapping with each other, which is seen in the similar KDE and GMM heatmaps. The two categories are quite distinct in the IN-12 Supervised model. This is captured in the each class lying in the other's low-likelihood region. The category separation is intermediate in the IN-12 DCL network. Though there is some

degree of separation between the classes compared to the Random initialization condition, it is much less than the IN-12 supervised.

### 3.3 INTER-CATEGORY DISTANCE

Given the probabilistic models of each category, we use two measures of category separation, one based on likelihood calculation and the other based on optimal transport.

**Log likelihood** We define the separation between two categories $c_1$ and $c_2$ as:
$$D_{LL}(c_1|c_2) = \mathcal{L}(X^{c_1}; p_{X^{c_1}}) - \mathcal{L}(X^{c_2}; p_{X^{c_1}})$$
where $\mathcal{L}(X^c; p_{X^{c_1}})$ is the average log-likelihood of the samples from class $c$ from the probabilistic model for class $c_1$. We calculate the log-likelihood based distance calculation for both KDE and GMM, yielding the KDE-LL and GMM-LL separation metrics.

**Earth Mover's Distance** The Earth Mover's Distance (EMD) is a metric for calculating distances between distributions and is the solution to the optimal transport problem. Formally, for two probability distributions $P$ and $Q$, it is defined as:
$$\text{EMD}(P, Q) = \inf_{\gamma \in \Pi(P,Q)} \mathbb{E}_{(x,y) \sim \gamma}[d(x, y)]$$
where $\Pi(P, Q)$ is the set of all distributions with marginals $P$ and $Q$. In this work, we only use the EMD metric with our GMM estimates. Delon & Desolneux (2020) proposed an approximation to the EMD for GMMs by constraining the set of coupling measures to gaussian mixture models. We utilize this method for calculating the EMD between GMMs. This produces the GMM-EMD metric for inter-category separation.

### 3.4 MEASURING SEPARABILITY

Measuring separability of a category $c$ requires aggregating the distance of $c$ from all other categories $c' \neq c$. Therefore, we define the separability $S(c)$ of a particular category $c \in C$ as:
$$S(c) = \frac{1}{|C| - 1} \sum_{c' \in C, c' \neq c} D(c|c')$$

Combined with our 3 metrics for inter-category distance, this produces 3 metrics for separability of a category $c$: $S_{KDE-LL}(c)$, $S_{GMM-LL}(c)$ and $S_{GMM-EMD}(c)$.

### 3.5 SEPARABILITY OF PRETRAINED FEATURES PREDICTS DOWNSTREAM JAN ACCURACY

We calculate separability for each category in the source and target datasets. If separability is important for downstream JAN accuracy, categories which have higher separability should show higher accuracy and vice-versa.

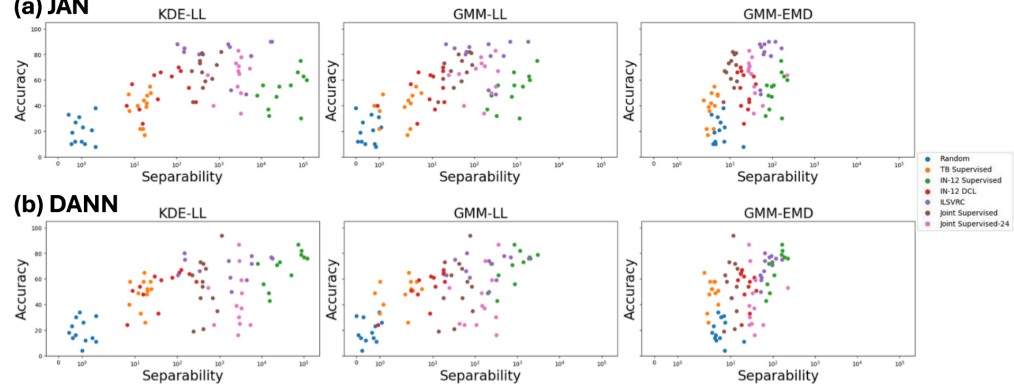

Figure 4: Scatter Plot of JAN **(a)** and DANN **(b)** accuracy on IN-12 test images with Separability metrics computed on the features for the IN-12 train images using different pretrained networks.

**Dependence of downstream performance on IN-12 Separability** Fig 4 shows the scatter plot between the IN-12 test accuracy after JAN/DANN training with the Separability of IN-12 train datapoints prior to DA. In Table 1 (first 2 rows), we calculate the correlation coefficient between

| Dataset | DA Method | KDE-LL | GMM-LL | GMM-EMD |
|---------|-----------|--------|--------|---------|
| IN-12 | JAN | **0.593 (p $<$ 0.001)** | **0.559 (p $<$ 0.001)** | **0.502 (p $<$ 0.001)** |
| | DANN | **0.595 (p $<$ 0.001)** | **0.56 (p $<$ 0.001)** | **0.51 (p $<$ 0.001)** |
| Toybox | JAN | **0.316 (p = 0.003)** | **0.375 (p $<$ 0.001)** | 0.123 (p = 0.265) |
| | DANN | 0.183 (p = 0.095) | **0.222 (p = 0.042)** | -0.065 (p = 0.555) |

Table 1: Correlation coefficient computed between downstream JAN/DANN accuracy and the logarithm of separability on the pretrained network (significant values (p $<$ 0.05) in **bold**).

downstream DA accuracy with each of the separability metrics on the IN-12 train datapoints on the different pretrained networks. The results strongly support our hypothesis: *JAN/DANN accuracy is strongly correlated with the separability of IN-12 train clusters with all 3 metrics.*

**Dependence of downstream performance on Toybox Separability**   Fig 8 in the Appendix shows the scatter plot between the IN-12 test accuracy after JAN/DANN training with the separability of Toybox train datapoints. Table 1 (last two rows) shows that the correlation between downstream JAN accuracy and the separability of Toybox train clusters is weaker with 2 metrics (KDE-LL, GMM-LL) and not of statistical significance with the GMM-EMD metric. On the other hand, DANN accuracy achieves significant correlation with only 1 metric (GMM-LL) and is not significant with the other two metrics.

# 4   LEARNING FEATURES BY ALIGNING DISTRIBUTION OF INTRA-DOMAIN PAIRWISE DISTANCES

Table 1 shows that separability of target categories in the feature space is beneficial for domain adaptation. However, without target labels, promoting target separability is difficult because no category information is available. To address this, *we propose two loss functions that maximizes the distribution similarity between source pairwise-image distances and target pairwise-image distances.*

We adopt a joint contrastive learning approach for learning from both Toybox and IN-12 datasets. This paradigm has recently been shown to be a competitive alternative to ILSVRC pretraining for several *instance-dominated* distribution shifts (Shen et al., 2022). Additionally, we add an MMD-based loss function that encourages the IN-12 dataset to have a similar distribution of pairwise distances as the Toybox dataset.

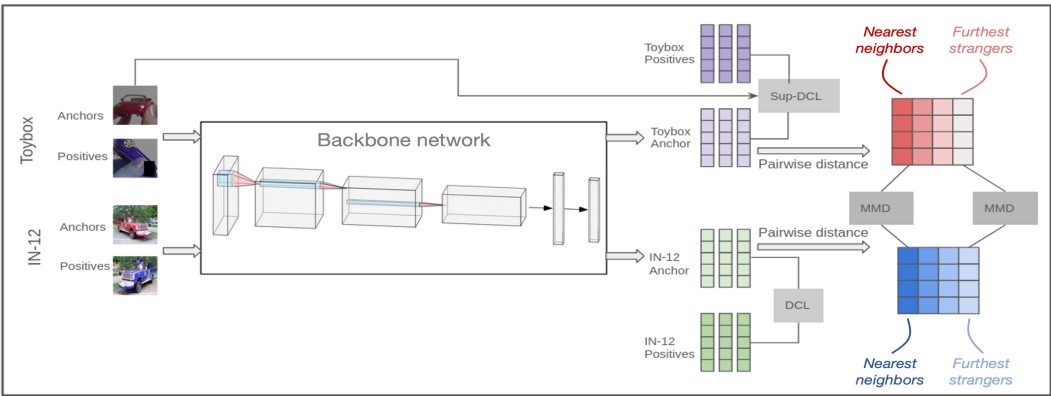

Figure 5: NAS-MMD for learning by aligning distribution of within-domain pairwise distances

## 4.1   JOINT CONTRASTIVE LEARNING

We use a joint contrastive learning framework for learning representations. In addition to being a powerful learning signal, contrastive loss functions are flexible and can be adapted to the presence of category labels. For IN-12, we use the Decoupled Contrastive Loss (DCL) (Yeh et al., 2022) signal to learn representations. DCL has been shown to outperform other contrastive methods with smaller batch sizes. For Toybox, we incorporate the category labels into the loss function. This is

done through two modifications to DCL: (1) All images within the same minibatch belonging to a particular category are considered as positive pairings for each other, and (2) We remove images from the same category from the batch negatives.

## 4.2 Learning features by aligning intra-domain distance distribution

Under the joint contrastive learning setting, we expect strong category separability to emerge for Toybox. *How can we leverage Toybox category separability to increase category separability in IN-12?*

**Pairwise-MMD**   We propose aligning the distribution of intra-domain pairwise image distances between the two domains. This requires two additional steps: (1) We calculate the intra-domain pairwise feature distance for each minibatch during training. (2) We use the MMD loss (Gretton et al., 2012) to minimize the distance between these two distributions. We call this the **Pairwise-MMD** loss.

**Neighbors-And-Strangers MMD (NAS-MMD)**   Within a minibatch, the number of across-class image distances largely outnumbers the number of within-class image distances. This might cause the MMD distance to weigh the across-class distances more strongly. To avoid this, we propose a variant, which we call Neighbors-And-Strangers MMD loss. For every image within a minibatch, we find its 3 nearest *neighbors* and 3 furthest *strangers* within the minibatch. We then apply the MMD loss separately on the *neighbors* distribution and on the *strangers* distribution between Toybox and IN-12. Fig 5 shows a schematic of the NAS-MMD method.

Empirically, we found that direct application of the MMD loss causes the Toybox distribution to shift to adapt to the IN-12 distribution. To prevent this, we apply the MMD loss and the NAS-MMD loss in an asymmetric manner: we restrict gradient flow through the source branch. This encourages the IN-12 features to shift to match the Toybox distribution.

## 4.3 Results

Table 2 shows the mean of 3 separate runs with each model. We see that addition of the Supervised DCL loss on Toybox leads to a small gain in performance for JAN and more benefits in the case of DANN. Recent work (Shen et al., 2022) has shown that jointly pretraining networks on source and target datasets yields strong DA results on other kinds of domain shifts. The performance gain, in our case, is much weaker and lags behind ImageNet pretraining. This suggests that the VI-Shift problem is different from other DA tasks in the literature.

Application of the pairwise-MMD and the NAS-MMD losses yields further benefits in performance. For JAN, either of those two variants outperforms the other models on 9 of the 12 categories. In case of DANN, the two variants outperform the baseline models on 10 of the 12 categories.

**Analysis of intra-domain feature distance distributions**   Fig 6 shows the histograms of pairwise image distances for Toybox and IN-12 datasets in the Joint contrastive training and the NAS-MMD settings respectively. The application of the NAS-MMD loss draws the within-class pairs in IN-12 to the left and increases the separation between the means of the two distributions.

| DA Method | Pre-training | airplane | car | helicopter | truck | cat | duck | giraffe | horse | ball | cup | mug | spoon | avg |
|---|---|---|---|---|---|---|---|---|---|---|---|---|---|---|
| JAN | IN-12 DCL | **63.00** | 68.67 | **66.00** | **69.00** | 58.33 | 60.67 | 44.67 | 53.00 | 38.00 | 37.00 | 38.67 | 10.67 | 50.64 |
| | + TB Sup-DCL | 58.00 | 66.00 | **66.00** | 56.67 | 54.33 | 57.33 | 71.00 | 53.33 | 36.67 | 31.00 | 54.67 | 19.00 | 52.00 |
| | + Pairwise-MMD | 58.33 | 68.33 | 65.0 | 51.0 | **62.0** | 63.33 | 79.33 | **56.00** | 34.33 | **38.00** | **58.00** | 31.67 | 55.45 |
| | + NAS-MMD | 55.67 | **70.67** | 64.00 | 60.00 | 60.00 | 58.33 | 76.33 | **56.00** | **42.67** | 35.67 | 51.33 | **50.00** | **56.72** |
| DANN | IN-12 DCL | 55.33 | 69.0 | 65.33 | 62.0 | 58.0 | 58.67 | 56.0 | 53.33 | 29.0 | **46.67** | 37.33 | 41.67 | 52.69 |
| | + TB Sup-DCL | 64.0 | **72.67** | 68.33 | 67.33 | **68.0** | 68.67 | 56.67 | 65.67 | 40.0 | 40.33 | 45.33 | 53.33 | 59.19 |
| | + Pairwise-MMD | 62.0 | 70.33 | 67.33 | **69.0** | 64.0 | **74.33** | 72.33 | **71.67** | **54.67** | 38.67 | 54.67 | **57.67** | **63.06** |
| | + NAS-MMD | **65.33** | **72.67** | **69.33** | 65.67 | 64.33 | 73.33 | **82.67** | 66.67 | 51.33 | 32.67 | **55.0** | 57.33 | 63.03 |

Table 2: Results (mean of 3 runs) showing the effectiveness of Pairwise-MMD and the NAS-MMD losses for JAN and DANN evaluation. These models outperform the baselines on 9 and 10 of the 12 categories for JAN and DANN respectively.

# 5 Related Work

**Domain Alignment**   Domain Alignment is a popular approach for addressing distribution shift in ML. Ben-David et al. (2010) showed that divergence between the two domains together with

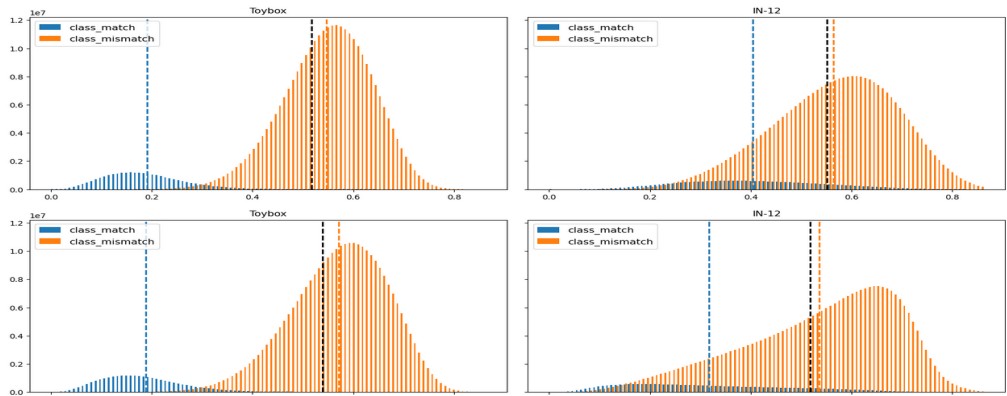

Figure 6: Histogram showing the distribution of pairwise distances in the Toybox and IN-12 datasets. **Top Row:** Histogram for IN-12 DCL + Toybox Sup-DCL training. While Toybox has two distinct components for within-class and across-class pairs, there is no such distinction in IN-12. **Bottom Row:** The across-class distribution also moves slightly to the left, but the gap between the two components (gap between dotted blue and orange lines) increases after NAS-MMD training.

empirical error on the source domain is a good approximation of the target error. One popular approach has been to use a distance metric based on the Maximum Mean discrepancy (Gretton et al., 2012); it is a distance measure between two distributions defined as the distance between the mean embeddings of the distribution in an RKHS. This metric has been used in several different variants to address problems of domain adaptation (Long et al., 2015; 2017; Ghifary et al., 2014; Kang et al., 2019) and domain generalization (Li et al., 2018).

**Data-driven approaches to cognitive science**   Our work is related to other recent work that leverage recent advances in deep learning to address important questions in the development of visual abilities in human infants. Bambach et al. (2018) demonstrated that CNNs learn better representations when they are trained on the visual experience of infants vs toddlers. In a similar vein, Stojanov et al. (2019) considered the problem of catastrophic forgetting in ML systems and showed that naturalistic patterns of repetition in an infant's visual experience significantly reduce the effect of catastrophic forgetting for visual object recognition. Orhan et al. (2020) looked at the problem of learning representations from videos from infants' play sessions and found that generic self-supervised learning methods can learn powerful high-level visual representations from this data. More recent work (Aubret et al., 2022) has shown that embodied visual experience presents strong signals for learning representations than models which have no access to these experiences.

## 6    CONCLUSION

In this work, we have introduced a novel distribution shift, called **VI-Shift**, over distributions of viewpoints and instances; this distribution is motivated by the visual experience of infants that drives category learning. We looked at this problem through the lens of domain adaptation in a developmentally plausible setting, i.e. without large-scale pretraining. We showed that two classic domain adaptation methods, JAN and DANN, underperform on this challenging task. Further, we sought to understand how separability of categories in the pretrained feature space affects downstream domain adaptation performance. To do this, we proposed 3 metrics for measuring category separability and showed that downstream JAN accuracy is strongly correlated with target cluster separability. To learn pretraining models more suited to the VI-Shift, we proposed two MMD-based methods for promoting category separability in the target dataset by matching its distribution of intra-domain pairwise image distances to that of the source domain. Our experiments show that this approach yields an improvement in the downstream accuracy with both JAN and DANN.

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

# A APPENDIX

## A.1 LIST OF IMAGENET SYNSETS USED FOR IN-12 DATASET

Fig 7 provides a list of the synsets used to compose the IN-12 dataset.

| Class | IN-12 candidate classes |
|---|---|
| Airplane | n02692086, n02691156, n04583620 |
| Ball | n02950943, n02882301, n03267113, n03445777, n02779435, n03982232, n02839351, n04254680, n03131967, n03742019, n04409515, n00474568, n02799071, n02778669 |
| Car | n04285008, n03268790, n02958343, n03870105, n02918964, n03828020, n04347119, n03770085, n04322801, n02960352, n02814533, n00449517, n04516354, n03141065, n04037443, n03079136 |
| Cat | n02126640, n02124313, n02982515, n02124484, n02123045, n02123242, n02122510, n02125081, n02123394, n02124075, n02122298, n02123478, n02123917, n02126787, n02121808, n02121620, n02122725, n02124623, n02126028, n02123597, n02125010 |
| Cup | n03733805, n03693707, n03216710, n07933799, n07930864, n04397452, n03147509, n03063073 |
| Duck | n01847978, n01847170, n01847407, n01846331, n01847253, n01852142, n01849157, n01852861, n01850873, n01852400, n01849863, n01849676, n01852671, n01854415, n01851375, n01851895, n01853195, n01851731 |
| Giraffe | n02439033 |
| Helicopter | n03512147, n04212467 |
| Horse | n02381460, n02387722, n02382948, n03539678, n02382338, n02379430, n02378541, n02377480, n02374451, n03061211, n02387254, n02381831, n02377291, n02386310, n02376918, n10186216, n00450335, n02377703, n04524142, n02387346, n02379183, n10185793, n00450070, n02379630 |
| Mug | n02824058, n03797390, n03063599 |
| Spoon | n04263502, n04284341, n04597913, n03180384, n04398688, n04284002, n03557270, n04350769, n04381073 |
| Truck | n04490091, n04461696, n03632852, n03417042, n04467665, n03256166, n03930630, n03345487, n03173929 |

Figure 7: Candidate classes from the ImageNet dataset used to create the IN-12 dataset

## A.2 HYPERPARAMETER TUNING DETAILS FOR JOINT SSL EXPERIMENTS

All models are trained for 150 epochs with an initial learning rate of 0.15 and a cosine decay schedule without restarts. Batch size was set to 256. We apply the Pairwise-MMD and the NAS-MMD losses starting from the 100th epoch. The relative weight of this loss increased from 0 to 1 during the last 50 epochs following a cosine schedule. The relative weight of the Toybox Supervised-DCL loss was set to 0.25. A larger value was found to reduce the effectiveness of the IN-12 DCL signal, while a smaller weight hampered separability of Toybox clusters.

## A.3 UMAP SETTINGS FOR METRIC CALCULATION

For calculating the metrics, we use the UMAP setting with n_neighbors set to 200, min_d set to 0.1 and using the Euclidean distance metric.

## A.4 LINEAR EVAL RESULTS

Table 3 shows our linear evaluation results.

## A.5 JAN AND DANN PERFORMANCE WITH DIFFERENT PRETRAINING SCHEMES

Table 4 shows our results using JAN and DANN with different pretraining schemes.

| Pre-training | Toybox | IN-12 | Linear Evaluation | |
| --- | --- | --- | --- | --- |
| | | | Toybox Test | IN-12 Test |
| None | ✗ | ✗ | 31.60 (0.97) | 36.42 (1.41) |
| Toybox Supervised | ✓ | ✗ | 76.45 (0.03) | 62.04 (0.41) |
| IN-12 DCL | ✗ | ✓ | 61.88 (0.25) | 81.50 (0.14) |
| IN-12 Supervised | ✗ | ✓ | 68.45 (0.27) | 86.55 (0.18) |
| Joint Supervised | ✓ | ✓ | 77.78 (0.02) | 84.84 (0.23) |
| Joint Supervised-24 | ✓ | ✓ | 80.02 (0.03) | 87.04 (0.05) |
| ILSVRC | ✓ | ✓ | 74.45 (0.23) | 90.88 (0.66) |

Table 3: JAN Accuracies with different pretraining schemes. Performance deteriorates in the absence of ImageNet pretraining.

| Pre-training | Infant-like DA framework | JAN Evaluation | | DANN evaluation | |
| --- | --- | --- | --- | --- | --- |
| | | Toybox Test | IN-12 Test | Toybox Test | IN-12 Test |
| None | Yes | 39.68 (1.12) | 19.83 (0.71) | 34.23 (2.48) | 17.36 (2.98) |
| Toybox Supervised | Yes | 64.90 (0.65) | 37.75 (1.30) | 64.78 (4.08) | 48.46 (1.00) |
| IN-12 DCL | Yes | 64.41 (0.99) | 50.64 (2.45) | 64.86 (1.42) | 52.29 (1.28) |
| IN-12 Supervised | No | 67.87 (0.62) | 53.21 (1.59) | 66.63 (0.26) | 66.34 (3.77) |
| Joint Supervised | No | 72.98 (0.96) | 68.59 (1.18) | 74.78 (0.36) | 51.46 (3.48) |
| Joint Supervised-24 | No | 76.81 (0.14) | 62.13 (0.42) | 76.58 (0.87) | 42.21 (1.24) |
| ILSVRC | No | 76.97 (0.22) | 79.09 (0.12) | 70.45 (1.76) | 69.67 (0.59) |

Table 4: JAN and DANN Accuracies with different pretraining schemes. Performance deteriorates in the absence of ImageNet pretraining.

### A.6 Scatter Plot of JAN/DANN accuracy on IN-12 test images with Toybox separability on different pretrained networks

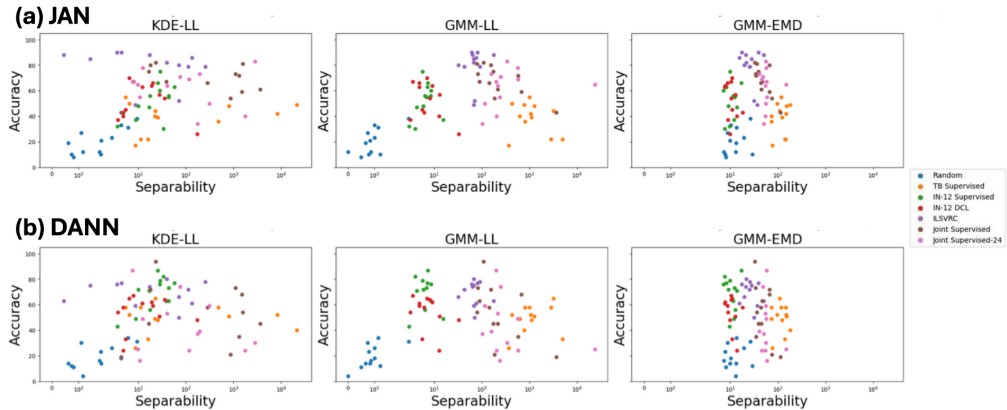

Figure 8: Scatter Plot of JAN **(a)** and DANN **(b)** accuracy on IN-12 test images with Separability metrics computed on the features for the Toybox train images using different pretrained networks.

### A.7 JAN training details

For the JAN evaluations, we initialize a bottleneck layer with 512 neurons. We follow the default training specifications provided in the JAN paper: the learning rate follows the schedule given by $\eta_p = 0.01(1 + 10p)^{-0.75}$, where p increases from 0 to 1 during training. The relative weight of the $l_{mmd}$ loss increases from 0 to 1 following $\lambda_p = \frac{2}{1+\exp(-10p)} - 1$. Each network is trained for 100 epochs with 100 minibatches per epoch using the SGD optimizer.

## A.8 DANN TRAINING DETAILS

For the DANN evaluations, the domain classifier has two hidden layers with 256 neurons. We follow the training specifications from *tllib* (Jiang et al., 2020): the learning rate follows the schedule given by $\eta_p = 0.01(1 + 10p)^{-0.75}$, where p increases from 0 to 1 during training. The relative weight of the $l_{dom}$ loss increases from 0 to 1 following $\lambda_p = \frac{2}{1+\exp(-10p)} - 1$. Each network is trained for 100 epochs with 100 minibatches per epoch using the SGD optimizer.

