# OpenReview forum: "Understanding Distribution Alignment Through Category Separability In An Infant-Inspired Domain Adaptation Task"
_ICLR.cc/2025/Conference — Submitted to ICLR 2025_

### Official Review · Reviewer_BHdw · 2024-10-27

**Soundness:** 3
**Presentation:** 3
**Contribution:** 2
**Rating:** 3
**Confidence:** 4

**Summary:**

This paper presents a question of how human and embodied visual agents learn object information with the tradeoff between object instances and object viewpoints. For this question, this work presents two kinds of data distribution: viewpoint-dominated (VD) distribution that is rich in the distribution of viewpoints; and also instance-dominated (ID) distribution that is rich in the number of different category instances.

To make a formal analysis of this problem, this work presents to study the DA problem from a VD dataset to an ID dataset with 12 categories. They found the the popular JAN model does not perform well. To this end, they propose 3 metrics to measure the category separability in the feature space and found that this is highly correlated with the JAN's performance. Finally, the work proposes a set of new loss functions that can improve the performance of JAN.

**Strengths:**

1. This work is motivated by an interesting problem.

In the Introduction, this paper presents a very interesting problem: is there a tradeoff between VD and ID data distribution when human or AI agents learn visual concepts? As mentioned in the paper, this problem has a lot of background support in psychology can could be potentially very interesting to explore in the area of visual learning.

2. This work includes a clear analysis and presentation.

This paper includes clear and comprehensive experiments about the properties of JAN, the proposed metrics, and loss functions. The visualizations are very clear and support the paper's arguments.

**Weaknesses:**

1. The relationship of human learning from VD vs ID data is not well connected with the DA problem.

Although this paper presents an interesting problem of the tradeoff between VD and ID data in visual concept learning, the main technical part of this paper focuses on doing DA from a VD dataset to an ID dataset. It is very vague to me why the problem of DA from VD to ID is related to the visual concept learning process from these data distributions. It seems to me that the technical part of this paper can be conducted on other DA problems, like from animation to real images, and also leads to the same conclusions. Therefore, it is unclear why VD and ID is important in this paper.

2. It is questionable whether it is necessary to study the problem assuming that the model is not trained on ImageNet.

The paper focuses on the performance of JAN without ImageNet pertaining; while the ImageNet pretrained version simply surpasses all the other models in this paper. It is unclear why it is important to ignore ImageNet pertaining, given that ImageNet pertaining is very common and widely used.

In other words, if the conclusion and proposed method only works for non-ImageNet pertaining model and yields worse performance than vanilla ImageNet pretrained ones, the technical application contribution of this paper is limited.

3. The focus on JAN needs more justification.

This paper's technical part centers around a popular DA method, JAN. I agree that JAN is a popular method and has been well studied, however, there are many other popular and newer DA methods (like MCD, for example) that claim to achieve better performance than JAN. This work should either prove that the conclusion and proposed method also works for newer methods or explain why only studying JAN is important and sufficient.

4. Unclear whether performance improvement of the proposed method is significant.

As shown in Table 3, the proposed method leads to performance improvement from 50.64 to 56.72. Although there is some improvement, it is unclear how significant this improvement is. After all, when compared with all other numbers in Table 1, 56.72 is still quite low and far from solving the problem.

**Questions:**

Please refer to the weakness section.

---

> ### Author Response · Authors · 2024-12-01
> **Response to Reviewer BHdw (part 1)**
>
> We would like to thank reviewer BHdw for the careful review and thoughtful feedback. We have made several changes to the manuscript, including additional evaluations using DANN and rewritten the Introduction section to convey our motivations and goals more clearly. Please see **global rebuttal** for additional details. Aside from that, the following are our response to your specific concerns and questions:
>
> ### 1. Focus on JAN
> Our choice of DA evaluation method was motivated by DA theory. DA theory [1] shows that low target domain error can be achieved by jointly optimizing the source error and the domain divergence. This result motivated our choice of JAN as the method. JAN uses an explicit alignment loss to optimize the domain divergence.
>
> We agree with the reviewers' comments that the choice of JAN may limit the significance of the results. To fix this, we have taken the reviewer's suggestion and added another DA method, Domain Adversarial Neural Network (DANN) in our analysis. DANN reduces the divergence between the two distributions by adopting an adversarial strategy and adheres to the results from DA theory. Our results using DANN strongly support our original conclusions using JAN, suggesting that our results apply to DA methods that are trained to reduce domain divergence. Please refer to the **global rebuttal** and the updated manuscript for the complete set of results.
>
>
> ### 2. Human learning from VD vs ID is not well connected to the DA problem
> Visual learning in infants is no doubt driven by both VD and ID datasets. However, our work is motivated by the nature of available labels for infants to learn from. Infants learn categories by linking heard words to objects that they see. Hence, learning is better when there are labels present for learning and there is no ambiguity about the referred object. Developmental psychology research shows that **(1)** parents talk more to their children during play sessions, and **(2)** referential ambiguity is significantly low when labels are heard because the objects the infant is holding dominates their field of view. On the other hand, in other situations, infant's views are cluttered and available labels are sparse and ambiguous.
>
> Our choice of domain adaptation as the suitable framework for studying this problem was motivated by these results. We have updated the **Introduction** section of the paper to include these details, state our goals more directly and include references from the infant development literature to support our choices. Please refer to the updated manuscript for these changes.
>
>
> ### 3. Why did we choose no ImageNet-pretraining?
> Almost all DA research relies on ImageNet-pretraining as the starting point. Pretraining on such a large dataset likely yields significant performance benefits. However, the goal of this work is developmental; we want to study *representation learning* in a developmentally plausible setting. Infants do not learn from massive, labeled datasets, so they cannot leverage the semantic knowledge that comes from ImageNet-pretraining while learning visual representations.
>
> We have updated the **Introduction** section of the paper to include these details regarding our motivation.
>
>
> ### 4. Performance of proposed models in comparison to ImageNet-pretrained models is low
> Yes, the proposed models perform worse than ImageNet-pretraining. Models that use ImageNet-pretraining use the considerable semantic and visual information that comes from training on ImageNet using millions of labels. However, such information is not present for an infant. They learn semantic knowledge by leveraging their visual experience, i.e. both **VD** and **ID** data. In this work, we investigate how we can learn such sophisticated semantic information directly from **VD** and **ID** datasets in a developmentally relevant setting.

---

> > ### Author Response · Authors · 2024-12-01
> > **Response to Reviewer BHdw (part 2)**
> >
> > ### 5. Technical contribution is limited
> > We strongly disagree with the reviewer's suggestion that technical contribution is only derived from performance comparison without considering the differences between and motivations behind different pieces of work. ImageNet-pretraining comes with extensive semantic information due to training on massive, labeled datasets, whereas the purpose of this study is to investigate mechanisms for such semantic knowledge to be learned from infant-like data distributions.
> >
> > We would request the reviewer to update their appraisal of our technical contributions given the distinctions of our work with other research using domain adaptation. We would like re-state our technical contributions for the reviewer's perusal:
> > 1. We investigated how separability of target clusters contribute to downstream accuracy using domain adaptation. We proposed 3 metrics that measure separability of clusters in the feature space. Further, we showed that separability of target clusters is strongly correlated to downstream DA accuracy using JAN/DANN.
> > 2. We seek to improve pretraining so that downstream DA evaluation performs better by increasing target separability during pretraining. To this end, we proposed two losses, **Pairwise-MMD** and **NAS-MMD**, application of which leads to improved DA performance.
> >
> >
> > ### 6. Unclear whether performance gain is significant
> >
> > We are working in a difficult learning setting, where the only signals can be derived from the learning task itself. Compared to ImageNet pretraining, this setting does not provide us with additional semantic information that we can leverage. We can only leverage regularities in the task definition to improve learned representations. **Pairwise-MMD** and **NAS-MMD** does this implicitly by exploiting the fact that Toybox clusters are well-separated in the feature space. We have replicated our results to include one additional DA method, DANN and all our results are supported by DANN evaluation.
> >
> > This is a new research direction and we are unaware of other methods that look at this problem. Thus, establishing the significance of the performance gains is difficult. However, a quick comparison with improvements reported in other DA papers can be useful. When the JAN paper was introduced, it reported a ~2% improvement over previous methods. In more recent work, the performance improvements are even less drastic: FixBi [2] reported improvement of <1% over existing methods and the same trend hold for CoVi [1].
> >
> > These comparisons make the performance improvements reported in our paper appear significant.
> >
> >
> > [1] Ben-David, Shai, et al. "A theory of learning from different domains." _Machine learning_ 79 (2010): 151-175.
> >
> > [2] Na, Jaemin, et al. "Fixbi: Bridging domain spaces for unsupervised domain adaptation." _Proceedings of the IEEE/CVF conference on computer vision and pattern recognition_. 2021.
> >
> > [3] Na, Jaemin, et al. "Contrastive vicinal space for unsupervised domain adaptation." _European Conference on Computer Vision_. Cham: Springer Nature Switzerland, 2022.

---

### Official Review · Reviewer_2pJc · 2024-10-27

**Soundness:** 2
**Presentation:** 3
**Contribution:** 2
**Rating:** 3
**Confidence:** 4

**Summary:**

This paper addresses the distribution shift problem inspired by infant visual learning. The authors propose metrics to measure category separability in feature space, demonstrating a strong correlation between target cluster separability and JAN accuracy. To improve performance, they introduce MMD-based loss functions that align the distribution of within-domain pairwise distances between source and target domains, leading to improved downstream performance.

**Strengths:**

1. An innovative perspective bridging cognitive science and machine learning through modeling infant visual learning patterns in domain adaptation. This interdisciplinary approach provides valuable insights into both fields while introducing a novel problem formulation.

2. The author conducted detailed empirical analysis of feature space characteristics through multiple metrics. The thorough evaluation demonstrates the relationship between category separability and model performance.

3. The author proposed a practical solution through the NAS-MMD loss function, which shows improvements across multiple categories.

**Weaknesses:**

1. The theoretical foundation (motivation) for the viewpoint-dominated to instance-dominated knowledge transfer is not clear. As the study performs classification based on images, different viewpoints capture the instance from different aspects and they (the viewpoint and instance) can’t be separated. And there are no relevant biological studies stating that infants transfer knowledge of objects from viewpoint to instance. The infants may be able to recognize objects mainly from instance-dominated images with limited viewpoint image support. So the motivation is not convincing and lacks evidence. The paper would benefit from additional citations or discussion of cognitive science literature supporting this specific learning progression.

2. The dataset construction methodology requires a more detailed explanation. While the paper mentions using ImageNet and MS-COCO for IN-12 construction, a more comprehensive description of the category selection criteria would improve reproducibility. For example, in Figure 1 (b), the categories of IN-12 should correspond to the Toybox dataset (as stated in line 120). It appears in the figure that the categories are mostly different.

3. The choice of JAN as the primary domain adaptation method needs stronger justification. A comparative analysis with other state-of-the-art domain adaptation distribution-related approaches (for example, [1][2]...) would better illustrate the findings. Also, the author should further elaborate on why chose JAN instead of DCL for distribution analysis? Why not directly make improvements on the Linear approach? The comparison of Linear evaluation in Table 1 lacks details (like network structure, parameter size, training process etc.)

[1] Distribution-Informed Neural Networks for Domain Adaptation Regression
[2] Distribution-Matching Embedding for Visual Domain Adaptation

4. The dimensional reduction technique selection (UMAP) would benefit from comparative analysis with alternatives like t-SNE and PCA. The rationale for removing outlier samples should be further elaborated.

5. The connection between the JAN-based distribution analysis and the DCL-based implementation could be better established. DCL may not share the same domain adaptation results as JAN.

6. The proposed NAS-MMD loss is a variant of MMD with limited contribution. The idea of moving neighbours closer and taking negative samples apart has been widely used in domain adaptation tasks.

7. The quality of the visual presentation in Figure 4 could be enhanced.

**Questions:**

Please refer to the weaknesses of the paper.

**Details Of Ethics Concerns:**

Nil

---

> ### Author Response · Authors · 2024-12-01
> **Response to Reviewer 2pJc (part 1)**
>
> We would like to thank reviewer 2pJC for the thoughtful review and constructive feedback. We have made several changes to the manuscript, including additional evaluations using DANN and rewritten the Introduction section to convey our motivations and goals more clearly. Please see **global rebuttal** for additional details. Aside from that, the following are our response to your specific concerns and questions:
>
>
> ### 1. Choice of JAN
> Our choice of DA evaluation method was motivated by DA theory. DA theory [1] shows that low target domain error can be achieved by jointly optimizing the source error and the domain divergence. This result motivated our choice of JAN as the method. JAN uses an explicit alignment loss to optimize the domain divergence.
>
> We agree with the reviewer's comments that the choice of JAN may limit the significance of the results. To fix this, we have taken the reviewer's suggestion and added another DA method, Domain Adversarial Neural Network (DANN) in our analysis. DANN reduces the divergence between the two distributions by adopting an adversarial strategy and adheres to the results from DA theory. Our results using DANN strongly support our original conclusions using JAN, suggesting that our results apply to DA methods that are trained to reduce domain divergence. Please refer to the **global rebuttal** and the updated manuscript for the complete set of results.
>
> ### 2. Theoretical foundation is unclear
> Our work was motivated by infant development literature that studies visual learning in infants. A long line of research has shown that object play is pervasive and rich learning opportunities for infants and such experiences are common across cultures, suggesting their importance for visual development. Additional literature shows that such experiences are beneficial for cognitive development of infants: infants who are more capable at motor skills show better object learning and 3d vision capabilities.
>
>
> We have updated the **Introduction** section of our work to include additional details about our motivations and the relevant infant development literature that supports our choices. Please refer to the **Introduction** section of the updated manuscript for these details.
>
>
> ### 3. Connection between JAN-based analysis and DCL-based implementation
> Our work focuses on the domain adaptation framework. We want to leverage DA methods in our study, but existing methods like JAN do not perform well without ImageNet pretraining (**Fig 1**). In our analysis of the feature space, we investigated whether target separability in the pretrained feature space is beneficial for evaluation using DA. Our results (**Table 1**) show that this is indeed true: pretrained networks that have higher target separability yield better DA performance using DANN and JAN. In our proposed method, we use this insight to propose two **cross-distribution** signals that, in addition to contrastive learning signals, aid pretrained networks that yield better accuracy under DA evaluation with DANN/JAN (**Fig 6** and **Table 2**).
>
> While our work uses the domain adaptation framework, there is a major difference between existing DA research and our work. While existing DA work use ImageNet pretraining as the starting point, we are proposing a method for pretraining that yields better DA results. We have updated the manuscript to convey that our method is intended for network pretraining and that we use DA methods like JAN and DANN as evaluation schemes for the pretrained models.
>
>
> ### 4. Details about the IN-12 dataset construction
> The images in the 2 datasets (Toybox and IN-12) are for illustration purpose only and are **not** row-matched in **Fig 1**. While each Toybox class has a corresponding category in IN-12, the first row in **Fig 1** does not show only images for IN-12 cars.
>
> For each of Toybox's 12 categories, we have 1600 images in IN-12 that are sourced from the ImageNet and MS-COCO datasets. For better understanding of our dataset construction, we have included the exact ImageNet synsets used for constructing IN-12 in the appendix (**Fig 7**).
>
> We will release the images that form the IN-12 dataset so it can be used by the research community. This should also aid reproduction of our results.

---

> > ### Author Response · Authors · 2024-12-01
> > **Response to Reviewer 2pJc (part 2)**
> >
> > ### 5. Why not improve results of linear probing?
> > Our goal in this work was to investigate how labeled **VD** images and unlabeled **ID** images can be leveraged to classify **ID** images in a developmentally plausible setting, i.e. without ImageNet pretraining. With this goal in mind, the more principled method of evaluation is to use a domain adaptation (DA) method. This led to our reliance on the DA evaluation as the metric for analysis. We did the linear evaluation to establish a *ceiling* for the DA methods and to demonstrate the gap between the accuracies between the two evaluation methods. In the overall setting, the linear evaluation is less relevant than the DA evaluation because linear evaluation requires that we use **ID** labels to train the linear classifier.
> >
> > We think that the manner in which the table was presented might be confusing, since the linear evaluation was, largely, only intended to make a point tangential to our main results in this work. To fix this, we have moved the linear evaluation results to the Appendix, and the main text of the paper only includes the results for the DA evaluation.
> >
> > ### 6. Proposed NAS-MMD is a variant of MMD with limited contributions
> > The novelty of this work is in the goal for applying the MMD loss. MMD is a metric that can be used for several different goals: in DA methods like JAN, it is used to align two distributions.
> > There are two major distinctions in our use of MMD compared to existing work:
> > - We use the MMD loss for a novel goal: to increase separability of the target features during training. Since we are operating in the domain adaptation setting, we can use labels to obtain *highly-separable* source clusters, while the target clusters lack this desired property. We use the MMD loss to increase the target separability by comparing with the distribution of the source features.
> > - Usually, MMD loss is directly applied on the feature space. This reduces the distance between the distributions. Instead, **Pairwise-MMD** and **NAS-MMD** match the distributions of intra-domain pairwise distances. This is intended to improve the *spread* of the target clusters in the feature space, and not overlay one distribution on another.
> >
> > [1] Ben-David, Shai, et al. "A theory of learning from different domains." _Machine learning_ 79 (2010): 151-175.

---

### Official Review · Reviewer_eUQf · 2024-11-04

**Soundness:** 2
**Presentation:** 3
**Contribution:** 2
**Rating:** 3
**Confidence:** 4

**Summary:**

This paper introduces VI-Shift, a new concept in distribution shift inspired by cognitive psychology and the way infants learn. The problem  of VI-Shift focuses on learning from viewpoint-dominated images and generalizing to instance-dominated images. To study this, the paper introduces two new benchmarks and formalizes the task as a domain adaptation (DA) problem. The authors also consider the popular DA method, Joint Adaptation Network (JAN), with ResNet backbones. Additionally, they propose two Maximum Mean Discrepancy (MMD)-based loss functions, Pairwise-MMD and Neighbors-And-Strangers-MMD (NAS-MMD), to improve model accuracy in this specific setting.

**Strengths:**

- The idea of focusing on VI-shit is novel and interesting, especially considering the connection between deep learning and cognitive science
- The two introduced benchmarks are interesting and can be used by other researchers to study this novel problem of VI-shift.
- The paper is well written and all the contributions of this work are well detailed.
- The paper has many contributions, ranging from introducing new datasets and task and proposing a deep architecture for the task.

**Weaknesses:**

-The paper primarily focuses on distribution alignment strategies, such as JAN, which is somewhat limiting given that various other domain adaptation (DA) paradigms could be explored to address the proposed VI-Shift problem. Specifically, adversarial domain adaptation methods, such as Domain-Adversarial Neural Networks (DANN) and Adversarial Discriminative Domain Adaptation (ADDA), could have been evaluated as potential solutions. Additionally, since the paper emphasizes contrastive learning, it would be valuable to examine how self-supervised learning approaches based on pretext tasks might perform in this context.
- The use of contrastive learning losses for domain adaptation is not novel, see e.g. [1]
- Since large Vision and Language Models (VLMs) are currently the mainstream it would have been important to show how multimodal models, such as CLIP, perform in this novel setting. Methods like CLIP have been already used for DA tasks, see e.g. [2]

[1] Da Costa, Victor G. Turrisi, et al. "Dual-head contrastive domain adaptation for video action recognition." Proceedings of the IEEE/CVF Winter Conference on Applications of Computer Vision. 2022.
[2] Singha, Mainak, et al. "Ad-clip: Adapting domains in prompt space using clip." Proceedings of the IEEE/CVF International Conference on Computer Vision. 2023

**Questions:**

- How other DA adaptation techniques (e.g. adversarial approaches) perform in this new setting?
- Could VLMs-based approaches be already effective to address the VI-shift problem.

---

> ### Author Response · Authors · 2024-12-01
> **Response to Reviewer eUQF**
>
> We would like to thank reviewer eUQF for the thoughtful feedback and insightful questions. We have made several changes to the manuscript, including additional evaluations using DANN and rewritten the Introduction section to convey our motivations and goals more clearly. Please see **global rebuttal** for additional details. Aside from that, the following are our response to your specific concerns and questions:
>
> ### 1. Focus on distribution alignment strategies like JAN is limiting
>
> Our choice of DA evaluation method was motivated by DA theory. DA theory [1] shows that low target domain error can be achieved by jointly optimizing the source error and the domain divergence. This result motivated our choice of JAN as the method. JAN uses an explicit alignment loss to optimize the domain divergence.
>
> We agree with the reviewer's comments that the choice of JAN may limit the significance of the results. To fix this, we have taken the reviewer's suggestion and added another DA method, Domain Adversarial Neural Network (DANN) in our analysis. DANN reduces the divergence between the two distributions by adopting an adversarial strategy and adheres to the results from DA theory. Our results using DANN strongly support our original conclusions using JAN, suggesting that our results apply to DA methods that are trained to reduce domain divergence. Please refer to the **global rebuttal** and the updated manuscript for the complete set of results.
>
> ### 2. Could VLMs be already effective to address the VI-Shift problem?
>
> While the use of VLMs is an interesting research direction, it is not consistent with our motivation. We are motivated by infant learning and our work is designed to study learning in a developmentally plausible setting. Infant category learning is driven by extensive experiences of handling a relatively small number of objects. This is why we focus on non-ImageNet pretraining in our work. While ImageNet-pretraining and other large-scale pretraining is effective for obtaining high-performing models, they are not suitable: infants do not learn from massive, labeled datasets.
>
> We have updated the **Introduction** section of the paper to convey our motivations and goals more clearly. We have explained why we choose to study the VI-Shift problem, why we adopt the domain adaptation framework for this study and why we focus on non-ImageNet pretraining evaluation. We have added references from the infant learning literature to support our choices. Please refer to the **global rebuttal** and the **Introduction** section of the updated manuscript for additional details.
>
> ### 3. The use of contrastive learning for domain adaptation is not novel
>
> While contrastive learning plays a part in our proposed methods, it is **not** our main contribution. Contrastive learning can be powerful and flexible for learning representations in different scenarios.
> Our contribution is the use of **cross-distribution** learning signals to aid representation learning. In this work, we have proposed two such **cross-distribution** signals, the **Pairwise-MMD** and **NAS-MMD** that try to improve cluster separability in the pretrained network. Our analysis showed that cluster separability is a desirable property of the pretrained network that drives DA performance, hence our design of these two signals.
> Our results show that addition of such **cross-distribution** signals during pretraining improves the performance of downstream DA evaluation (**Table 2**). The structure of the feature space is also improved by the application of these additional signals (**Fig 6**).
>
> While we use DA methods for evaluation, our goal in proposing the **Pairwise-MMD** and the **NAS-MMD** signals was in designing better pretraining models that existing DA methods can leverage. While existing DA methods have used contrastive learning, our goals are different. Our proposed signals are not directly for domain adaptation, but yield pretrained feature spaces that are better suited for DA methods that are trained to reduce the domain divergence.
>
> ### 4. Other self-supervised signals
>
> There are a lot of SSL methods with their own merits and demerits. However, our work is complementary to this line of research. SSL methods learn from a single distribution; in this work, we have investigate learning from the **VD** and **ID** distributions. In this setting, we can apply SSL signals to learn from both distributions, but we show that adding **cross-distribution** signals improves the quality of the learned representations. Other SSL methods are not directly comparable in our case, since they do not use such **cross-distribution** signals.

---

### Official Review · Reviewer_rPZQ · 2024-11-06

**Soundness:** 2
**Presentation:** 2
**Contribution:** 2
**Rating:** 5
**Confidence:** 4

**Summary:**

*Problem*: Domain adaption in a setting where the labelled (source) dataset, Toybox12, has images of a small number of object instances from 12 categories, but from many viewpoints. The unlabelled (target) dataset is IN-12, containing natural images with many instances of the same categories sourced from ImageNet/MSCOCO.

*Methodology*: Pretrain features with a variety of techniques, from supervised to self-supervised, and then finetune the network with a domain adaptation method (JAN). Measure the final classification accuracy as the evaluation metric.

*Analysis*: The authors develop three metrics to quantify the separability of class clusters in the pretrained feature space, and draw correlations between these metrics and downstream JAN accuracy. Based on these findings, authors develop a new metric which performs better than DCL constrastive learning initialisation.

**Strengths:**

* I find the problem tackled in this paper to be very interesting and worthy of further study. The problem setting mimics the way in which infants learn about object categories - wherein only a few object instances are available to the infant, but the infant has the ability to play with the object and look at it from a number of viewpoints.
* The dataset contribution seems interesting, with the proposed dataset extending the Visda-2017c challenge with real videos and test instances, rather than rendered 3D objects for training.
* The proposed method provides a solid boost over standard DCL contrastive learning baseline, improving performance from 50.6 to 56.7.

**Weaknesses:**

* The main weakness for me is the primary evaluation methodology. Specifically, authors use JAN finetuning and accuracy as a metric by which to evaluate pretrained features. I can see two issues with this: (i) in many cases, simple linear probing seems to substantially outperform JAN evaluation (Table 1). I cannot see why linear probing is not used as the primary metric. Given this, it is unclear to what extent the findings are an artefact of JAN failure modes. (ii) In a similar vein, if a domain adaptation method is to be used as the evaluation metric, why not use a more modern approach, e.g (FixBi, Na et al. 2021, CoVi Na et al. 2021).
    * Perhaps I have misunderstood something here.
* The proposed metrics to measure feature space separability seem a little underwhelming to me, with a maximum correlation with final accuracy of only 0.5. This indicates that either other metrics might be better (e.g simple k-means/k-NN accuracy) or that cluster separation of pretrained features is not actually linked to downstream JAN accuracy.
* The authors propose a new contrastive learning technique without trying extensive baselines over simple DCL. For instance, supervised contrastive learning is quite similar in spirit to the proposed method (Supervised Contrastive Learning, Khosla et al).

**Questions:**

* Did the authors try other self-supervised or contrastive baselines? (e.g MAE, Supervised Contrastive Learning, DINO etc)
* Did the authors try other class separability metrics, like k-NN or k-Means?

---

> ### Author Response · Authors · 2024-12-01
> **Response to reviewer rPZQ (part 1)**
>
> We would like to thank reviewer rPZQ for the review, questions raised and the constructive feedback. We have made several changes to the manuscript, including additional evaluations using DANN and rewritten the Introduction section to convey our motivations and goals more clearly. Please see **global rebuttal** for additional details. Aside from that, the following are our response to your specific concerns and questions:
>
>
> ### 1. Additional experiments with DANN evaluation
>
> Our choice of DA evaluation method was motivated by DA theory. DA theory [1] shows that low target domain error can be achieved by jointly optimizing the source error and the domain divergence. This result motivated our choice of JAN as the method. JAN uses an explicit alignment loss to optimize the domain divergence.
>
> We agree with the reviewer's comments that the choice of JAN may limit the significance of the results. To fix this, we have taken the reviewer's suggestion and added another DA method, Domain Adversarial Neural Network (DANN) in our analysis. DANN reduces the divergence between the two distributions by adopting an adversarial strategy and adheres to the results from DA theory. Our results using DANN strongly support our original conclusions using JAN, suggesting that our results apply to DA methods that are trained to reduce domain divergence. Please refer to the **global rebuttal** and the updated manuscript for the complete set of results.
>
>
> ### 2. Why not use linear probing as the metric?
>
> Our goal in this work was to investigate how labeled **VD** images and unlabeled **ID** images can be leveraged to classify **ID** images in a developmentally plausible setting, i.e. without ImageNet pretraining. With this goal in mind, the more principled method of evaluation is to use a domain adaptation (DA) method. This led to our reliance on the DA evaluation as the metric for analysis. We did the linear evaluation to establish a *ceiling* for the DA methods and to demonstrate the gap between the accuracies between the two evaluation methods. In the overall setting, the linear evaluation is less relevant than the DA evaluation because linear evaluation requires that we use **ID** labels to train the linear classifier.
>
> We think that the manner in which the table was presented might be confusing, since the linear evaluation was, largely, only intended to make a point tangential to our main results in this work. To fix this, we have moved the linear evaluation results to the Appendix, and the main text of the paper only includes the results for the DA evaluation.
>
> ### 3. Why not SOTA domain adaptation methods?
>
> We were motivated by results in the theory of DA in choosing the method of choice. DA theory [1] suggests that low error on the target domain can be achieved by jointly optimizing the source error and the domain divergence. For the work in the paper, we chose JAN, because JAN directly adheres to these suggestions. Additionally, JAN also performs competitively on many benchmarks.
>
> Recent methods like FixBi [2] and CoVi [3] rely on sophisticated pseudo-label based training and mixup-based augmentations; while they do perform better than JAN on DA benchmarks, their relation with DA theory is less explored. Additionally, the effectiveness of these models is highly dependent on obtaining reliable labels from the pretrained network directly
>
> To make our results stronger and to ensure that our results apply to methods that follow DA theory and not just JAN, we have added another DA method, Domain Adversarial Neural Networks (DANN) in our analysis. We have redone all evaluations with DANN and these results strongly support our original conclusions. Please refer to the **Global Rebuttal** and the revised manuscript for detailed description of these additional results.
>
> ### 4. Correlation between separability and downstream accuracy is 0.5, might not be significant
>
> We strongly disagree with the reviewer about the significance and the strength of the correlation. A correlation of 0.5 is typically considered quite high in statistical analysis, especially since we have included *p-values* along with the correlation values. If the cluster separability in pretrained features is not related to downstream accuracy, we would see correlation values close to 0 with a high *p-value*. This is indeed the case when we try to relate the separability of the source dataset with the downstream DA accuracy with two of our metrics. But our claims in the paper are only about the correlations between the separability of **target** clusters and, as shown in **Table 1**, these values are high and also statistically significant.

---

> > ### Author Response · Authors · 2024-12-01
> > **Response to Reviewer rPZQ (part 2)**
> >
> > ### 5. Other metrics like k-NN and k-means
> >
> > For our work, we strongly feel that simpler techniques like k-NN and k-means are either incompatible with the objective of this work or weaker than the proposed metrics.
> >
> > To use k-NN as an accuracy metric, we need to access **ID** labels to obtain predictions for the other **ID** images. This runs counter to the goal of this work, which is to use labels from only **VD** images to predict labels for **ID** images.
> >
> > The calculation of the separability values requires two steps: **(1)** obtaining cluster assignments for each image, and **(2)** calculating separability between each pair of clusters. K-means would give us a way to make assign individual images to clusters, but this is inherently weaker than using labels directly to determine the clusters: using image labels provides perfect information, whereas k-Means would be noisy. Because we are analyzing the feature space and not making predictions, directly using labels for both **VD** and **ID** images does not contradict our goal. Either way, after we get cluster assignments, we would still be left with the problem of calculating the separability between clusters, which is the use case for our proposed metrics, such as **KDE-LL**, **GMM-LL** and **GMM-EMD**.
> >
> > ### 6. Contrastive baselines
> >
> > Our contribution in this work is not the contrastive methods, but the application of **cross-distribution** learning signals, such as **Pairwise-MMD** and **NAS-MMD**.
> >
> > Our task framework requires that we use **unlabeled ID** images and **labeled VD** images for pretraining. Self-supervised learning (SSL) is a natural framework for learning features in this setting, since they are flexible about the learning signal: on **ID** images, we can use regular SSL signals, whereas for **VD** images, we can incorporate the labels into the learning signal. The latter is similar to Supervised Contrastive learning. Instead, we opt for DCL as the base method, since it performs better with smaller batch sizes. However, our contribution is **not** the contrastive learning methods.
> >
> > Instead, our contribution is the contribution of **cross-distribution** signals that augment the contrastive learning signals. These signals are complementary to the underlying SSL signals used. In this work, we proposed two such signals, **Pairwise-MMD** and **NAS-MMD** with the aim of improving the separability of target clusters during pretraining. As our results show (**Table 2**), these signals do show an improvement of just applying contrastive learning signals on the two datasets: accuracies on the target dataset are higher using both JAN and DANN. Additionally, our analysis of the pretrained feature space (**Fig 6**) shows that these signals improve the structure of the feature space as well: the distributions of within-class and across-class distances for the target dataset are more separated when such **cross-distribution** signals are added.
> >
> >
> >
> > [1] Ben-David, Shai, et al. "A theory of learning from different domains." _Machine learning_ 79 (2010): 151-175.
> >
> > [2] Na, Jaemin, et al. "Fixbi: Bridging domain spaces for unsupervised domain adaptation." _Proceedings of the IEEE/CVF conference on computer vision and pattern recognition_. 2021.
> >
> > [3] Na, Jaemin, et al. "Contrastive vicinal space for unsupervised domain adaptation." _European Conference on Computer Vision_. Cham: Springer Nature Switzerland, 2022.

---

### Author Response · Authors · 2024-12-01
**Global Response**

# Global Response
We would like to thank all reviewers for their comments and constructive feedback about our submission. We have carefully considered all suggestions and comments made by the four reviewers.  Based on these comments, we have performed additional computational experiments. These experiments **strongly** support our original conclusions. In addition, reviewers had concerns about the background and motivation behind the task setup considered in the current work. We have updated the manuscript to respond to these concerns.

## Background and Motivation:
Reviewers *BHdw* and *2pJc* had concerns about the task setup and motivation behind the current work. We have updated the **Introduction** section to clarify these concerns by **(1)** stating our goals and motivations more directly,  and **(2)** by including additional references from the infant development literature that support our task setup.


## Additional Experiments and Results
All reviewers had concerns about our choice of one domain adaptation method, JAN, in our analysis. To demonstrate the generality of our results, we have added an additional method of domain adaptation in our analysis, *Domain Adversarial Neural Network (DANN)* [1]. The updated manuscript includes additional material introducing *DANN* (**section 2**) and additional results using *DANN* (**sections 2, 3, 4**). These additional results **support** our original hypotheses. For the ease of the reviewers, we want to state these additional results and their similarity to the results in the original submission:
1. *Evaluation without ImageNet pretraining:* In the original submission, we saw that **JAN** performance deteriorates in developmentally plausible training situations, i.e. without ImageNet pretraining. Our new results with **DANN** also shows similar deterioration in performance with developmentally relevant pretraining. For the complete results, refer to **Fig 2** in the revised manuscript and **Table 4** in the Appendix.
2. *Target separability in pretrained feature space predicts downstream DA accuracy:* We had hypothesized that separability of target features in the pretrained feature space is beneficial for downstream DA accuracy. While our original results had only considered JAN as the DA method, we have now included DANN into this analysis as well. The results are similar. Using both methods, we see strong and significant correlation between separability in pretrained feature space and downstream DA accuracy. Please see **Fig 4** and **Table 1** in the revised manuscript for these results.
3. *Learning by aligning pairwise distances:* To promote target separability, we had incorporated two loss functions, **Pairwise-MMD** and **NAS-MMD** during pretraining along with contrastive loss-based learning signals. In the *original* submission, we showed that these additional loss functions make the pretrained network better suited for JAN evaluation - there was a modest improvement in JAN evaluation accuracy on the target dataset, IN-12. In the **revised** manuscript, we have added DANN evaluation for these pretrained models as well. Our results show a similar improvement in DANN evaluation accuracy when using these additional loss functions. The complete results are presented in **Table 2** in the revised manuscript.

**References**
[1] Ganin, Yaroslav, et al. "Domain-adversarial training of neural networks." *Journal of machine learning research* 17.59 (2016): 1-35.

## Other changes
Reviewers *rPZQ* and *2pJc* questioned why we did not improve upon the performance of the linear evaluation results. The linear evaluation results require us to use the target labels from IN-12 to train the linear classifier. This contradicts with our task setup following the domain adaptation paradigm because we require that IN-12 labels not be used in training the network in any form. The linear evaluation results were included to show that DA methods fail to find good classifiers even when they exist. However, we understand the confusion that reviewers had in looking at those results. To remedy this, we have moved the linear evaluation results to the Appendix (**Table 3**) and the main paper includes an additional figure (**Fig 2**) that show the results for the DA evaluation only.

In addition to this global rebuttal, we have a response to specific questions that each of the reviewers raised as comments to their responses. We look forward to hearing reviewers' feedback regarding these updates in the revised manuscript.

---

### Meta-Review · Area_Chair_76vA · 2024-12-23

**Metareview:**

This paper considers the problem of domain adaptation in an interesting setting, drawing inspiration from cognitive psychology and early learning behaviors. It specifically addresses the transition from viewpoint-centric images to instance-focused images. The study establishes two benchmarks, framing the issue as a domain adaptation challenge. Evaluating the popular JAN approach with ResNet backbones, the authors introduce Pairwise-MMD and NAS-MMD loss functions based on Maximum Mean Discrepancy to boost model accuracy within this unique scenario.

**Additional Comments On Reviewer Discussion:**

The majority of the reviewers appreciate that the task is interesting. Meanwhile, the majority of the reviewers have also expressed significant concerns no the readiness of this work. The final ratings of the submission are 3x reject and 1x borderline reject. The AC does not spot any strong reason to turn down the reviewers' recommendation.

---

### Decision · Program_Chairs · 2025-01-22

Reject